# TriBERT: Full-body Human-centric Audio-visual Representation Learning for Visual Sound Separation

**Tanzila Rahman**[1,3]    **Mengyu Yang**[2,3]    **Leonid Sigal**[1,3,4]
[1]University of British Columbia    [2]University of Toronto
[3]Vector Institute for AI    [4]Canada CIFAR AI Chair
trahman8@cs.ubc.ca, my.yang@mail.utoronto.ca, lsigal@cs.ubc.ca

## Abstract

The recent success of transformer models in language, such as BERT, has motivated the use of such architectures for multi-modal feature learning and tasks. However, most multi-modal variants (e.g., ViLBERT) have limited themselves to visual-linguistic data. Relatively few have explored its use in audio-visual modalities, and none, to our knowledge, illustrate them in the context of granular audio-visual detection or segmentation tasks such as sound source separation and localization. In this work, we introduce TriBERT – a transformer-based architecture, inspired by ViLBERT, which enables contextual feature learning across three modalities: vision, pose, and audio, with the use of flexible co-attention. The use of pose keypoints is inspired by recent works that illustrate that such representations can significantly boost performance in many audio-visual scenarios where often one or more persons are responsible for the sound explicitly (e.g., talking) or implicitly (e.g., sound produced as a function of human manipulating an object). From a technical perspective, as part of the TriBERT architecture, we introduce a learned visual tokenization scheme based on spatial attention and leverage weak-supervision to allow granular cross-modal interactions for visual and pose modalities. Further, we supplement learning with sound-source separation loss formulated across all three streams. We pre-train our model on the large MUSIC21 dataset and demonstrate improved performance in audio-visual sound source separation on that dataset as well as other datasets through fine-tuning. In addition, we show that the learned TriBERT representations are generic and significantly improve performance on other audio-visual tasks such as cross-modal audio-visual-pose retrieval by as much as 66.7% in top-1 accuracy.

## 1 Introduction

Multi-modal audio-visual learning [57], which explores and leverages the relationship between visual and auditory modalities, has started to emerge as an important sub-field of machine learning and computer vision. Examples of typical tasks include: *audio-visual separation and localization*, where the goal is to segment sounds produced by individual objects in an audio and/or to localize those objects in a visual scene [15, 16, 42, 55]; and *audio-visual correspondence*, where the goal is often audio-visual retrieval [23, 47, 53]. Notably, some of the most recent audio-visual methods [15] leverage human pose keypoints, or landmarks, as an intermediate or contextual representation. This tends to improve the overall performance of sound separation, as pose and motion are important cues for characterising both the type of instrument being played and, potentially, over time, the rhythm of the individual piece [15]. It can also serve as an intermediate representation when generating video from acoustic signals [8, 44] for example.

35th Conference on Neural Information Processing Systems (NeurIPS 2021).

Most of the existing architectures tend to extract features from the necessary modalities using pre-trained backbones (*e.g.*, CNNs applied to video frames [55], object regions [16], and audio spectrograms; and/or graph CNN for human pose [15]) and then construct problem-specific architectures that often utilize simple late fusion for cross-modal integration in decoding (*e.g.*, to produce spectrogram masks [15, 16, 55]). This is contrary to current trends in other multi-modal problem domains, where over the past few years, approaches have largely consolidated around generic multi-modal feature learning architectures that are task agnostic to produce contextualized feature representations and then fine-tune those representations to a variety of tasks (*e.g.*, visual question answering (VQA) or reasoning (VCR)) and datasets. Examples of such architectures include ViLBERT [33], VL-BERT [46], and Unicoder-VL [31], all designed specifically for visual-linguistic tasks.

Audio-visual representation learning has, in comparison, received much less attention. Most prior works [51] assume a single sound source per video and rely on audio-visual alignment objectives. Exceptions include [39], which relies on proposal mechanisms and multiple-instance learning [49] or co-clustering [25]. These approaches tend to integrate multi-modal features extracted using pre-trained feature extractors (*e.g.*, CNNs) at a somewhat shallow level. The very recent variants [6, 28, 35] leverage transformers for audio-visual representation learning through simple classification [6] and self-supervised [28] or contrastive [35] learning objectives while only illustrating performance on video-level audio-visual action classification. To the best of our knowledge, no audio-visual representation learning approach to date has explored pose as one of the constituent modalities; nor has shown that feature integration and contextualization at a hierarchy of levels, as is the case for BERT-like architectures, can lead to improvements on granular audio-visual tasks such as audio-visual sound source separation.

To address the aforementioned limitations, we formulate a human-centric audio-visual representation learning architecture, inspired by ViLBERT [33] and other transformer-based designs, with an explicit goal of improving the state-of-the-art in audio-visual sound source separation. Our transformer model takes three streams of information: video, audio, and (pose) keypoints and co-attends among those three modalities to arrive at enriched representations that can then be used for the final audio-visual sound separation task. We illustrate that these representations are general and also improve performance on other auxiliary tasks (*e.g.*, forms of cross-modal audio-visual-pose retrieval). From a technical perspective, unlike ViLBERT and others, our model does not rely on global frame-wise features nor an external proposal mechanism. Instead, we leverage a learned attention to form visual tokens, akin to [42], and leverage weakly supervised objectives that avoid single sound-source assumptions for learning. In addition, we introduce spectrogram mask prediction as one of our pre-training tasks to enable the network to better learn task-specific contextualized features.

**Contributions:** Foremost, we introduce a tri-modal VilBERT-inspired model, which we call TriBERT, that co-attends among visual, pose keypoint, and audio modalities to produce highly contextualized representations. We show that these representations, obtained by optimizing the model with respect to uni-modal (weakly-supervised) classification and sound separation pretraining objectives, produce features that improve audio-visual sound source separation at large and also work well on other downstream tasks. Further, to avoid reliance on the image proposal mechanisms, we formulate tokenization in the image stream in terms of learned attentional pooling, which is learned jointly. This alleviates the need for externally trained detection mechanisms, such as Faster R-CNN and variants. We illustrate competitive performance on a number of granular audio-visual tasks both by using the TriBERT model directly, using it as a feature extractor, or by fine-tuning it.

## 2 Related works

**Audio-visual Tasks.** There exists a close relationship between visual scenes and the sounds that they produce. This relationship has been leveraged to complete various audio-visual tasks. Based on [57]'s survey of audio-visual deep learning, these tasks can be categorized into four subfields, three of which are addressed in this paper and described in the following three subsections.

**Audio-visual Sound Source Separation and Localization.** Sound source separation and the related task of sound source localization have been studied quite extensively. Previous works studying separation, also known as the *cocktail party problem* [19], leverage multi-modal audio-visual information [11, 14] to help improve performance with respect to their audio-only counterparts [26, 34]. Examples include learning correlations between optical flow and masked frequencies [9, 13], using

graphical models [21], detecting salient motion signals that correspond to audio events [30, 40], and extracting pose keypoints to model human movements [15]. A close connection between separation and localization has also been illustrated [40, 43, 55, 56]. For example, [16, 42] both formulate the task as one of auditory and visual co-segmentation, either with pre-trained object regions obtained by the detector [16] or directly from the image [42]. All of these approaches contain highly specialized architectures with custom fusion schemes. We aim to leverage the flexibility of transformer models to create generalized multi-modal representations that improve on audio-visual tasks.

**Audio-visual Representation Learning.** The goal is typically to learn *aligned* representations. The quality of these representations has been shown to greatly impact the overall performance of tasks downstream [4]. A common strategy for representation learning is to introduce a proxy task. In the audio-visual space, past works [1, 2, 38] have trained networks by having them watch and listen to a large amount of unlabeled videos containing both positive samples of matching audio and visual pairs and negative samples of mismatched pairs; the proxy task is binary classification of whether or not the audio and visual match each other. Other proxy tasks include determining whether or not an audio-visual pair is time synchronized [27]; and [29] uses a classification task to identify the correct visual clip or audio stream from a set with negative samples. However, these works rely on the assumption that only one main sound source occurs at a time and everything else is background noise. Our model uses a weakly supervised proxy objective to learn representations for multiple sources of sound (two in experiments) occurring simultaneously and also learns to incorporate pose features.

**Audio-visual Correspondence Learning.** One of the fundamental tasks in correspondence learning related to our work is cross-modality retrieval. Most prior works focus on audio-visual retrieval [24, 36, 48] and propose learning a joint embedding space where both modalities can be mapped to. In this space, semantically related embeddings are close to each other and thus retrieval can be performed by selecting the closest embedding to the query from the alternate modality. In our work, we demonstrate that enhanced feature representations obtained by our pretrained model capture aligned semantics and lead to much better cross-modal retrieval than baseline representations.

**Visiolinguistic Representation Learning.** Our model is inspired by the recent successes of visiolinguistic representations. Most such approaches leverage a combination of uni-modal and cross-modal transformer modules to pre-train generic visiolinguistic representations on masked language and/or multi-modal alignment tasks. For example, [33] proposes separate streams for each modality that communicate with each other through co-attention, while [46] uses a single-stream model that takes both visual and linguistic embeddings as input. In our work, we also leverage co-attention modules to learn joint representations between audio, pose, and vision modalities. However, in addition to extending co-attention, we also focus on reformulating image tokenization and demonstrate the ability to learn with weakly-supervised classification objectives as opposed to masked token predictions.

## 3   Approach

We introduce TriBERT, a network that learns a joint representation of three modalities: vision, pose, and audio. We briefly review ViLBERT, the architecture that inspired TriBERT, in Section 3.1. We then describe our TriBERT architecture in Section 3.2, including pretraining tasks and objectives.

### 3.1   Reviewing Vision-and-Language BERT (ViLBERT)

Motivated by the recent success of the BERT architecture for transfer learning in language modeling, Lu *et al.* [33] proposed ViLBERT to represent text and visual content jointly. ViLBERT is a two-stream model for image regions and text segments. Each stream is similar to the BERT architecture, containing a series of transformer blocks (TRM) [50]. Given an image $I$ with corresponding regions-of-interest (RoIs) or bounding boxes $v_0, v_1, ...v_N$ and an input sentence $S$ with word tokens $w_0, w_1, ...w_T$, the final output representations are $h_{v0}, h_{v1}, ..., h_{vN}$ and $h_{w0}, h_{w1}, ..., h_{wT}$ for the visual and linguistic features, respectively. To exchange information between the two modalities, the authors introduced a co-attentional transformer layer which computes query ($Q$), key ($K$), and value ($V$) pairs like a standard transformer block. The keys and values from each modality are then fed to the multi-headed attention block of the other modality. The attention block in each stream generates attention-pooled features conditioned on the other modality and outputs a multi-modal joint representation which outperforms single-stream models across multiple vision-and-language tasks.

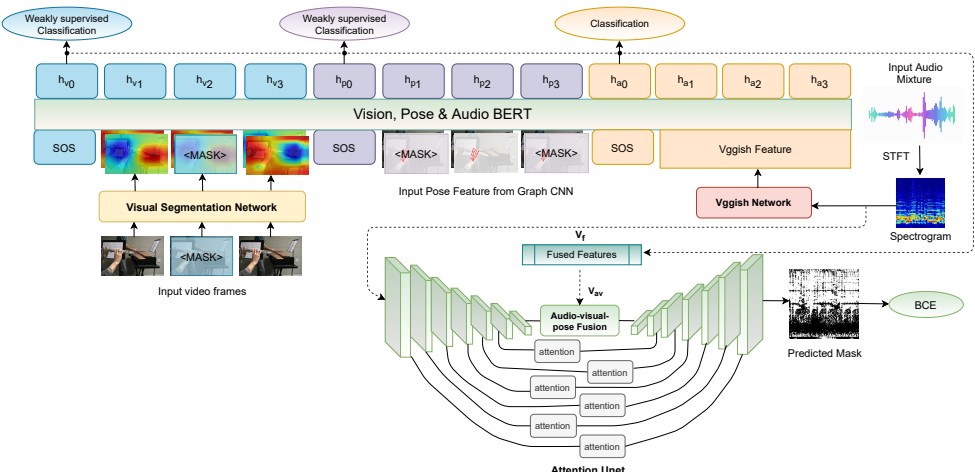

Figure 1: **Our TriBERT Architecture.** We train TriBERT on the MUSIC21 dataset under two training tasks: (1) Classification and (2) Sound separation. We introduce an end-to-end segmentation network for visual embeddings which takes consecutive RGB frames as input and outputs detected object features to feed into vision BERT. Following [15], we use graph CNN to generate pose embeddings as input to pose BERT. For audio, we consider the mixed spectrogram of two audio sources and use a VGGish network to generate audio embeddings to feed into audio BERT. We use classification loss to train individual modalities. Due to the lack of annotation for individual objects, we use a weakly supervised classification loss to train TriBERT for the vision and pose streams. Following prior works on sound separation, we utilize attention U-net [37], which takes a mixed audio spectrogram as input and predicts a spectrogram mask guided by audio-visual-pose features.

## 3.2 TriBERT Architecture

The architecture of our proposed TriBERT network is illustrated in Figure 1. Inspired by the recent success of ViLBERT in the vision-and-language domain, we modify its architecture to a three-stream network for vision, pose, and audio. Similar to ViLBERT [33], we use a bi-directional Transformer encoder [50] as the backbone network. However, TriBERT also introduces integral components that differentiate its architectural design. First, instead of using bounding box visual features generated by a pre-trained object detector [33] or CNN feature columns [7], TriBERT uses a jointly trained weakly supervised *visual segmentation network*. Our end-to-end segmentation network takes a sequence of consecutive frames to detect and segment objects, and the corresponding features are pooled and fed as tokens to the visual stream. Second, the pose tokens are characterized by per-person keypoints encoded using a Graph CNN, and the audio token is produced by the VGGish Network [22] applied to an audio spectrogram. All three types of tokens form the input to TriBERT, which refines them using tri-modal co-attention to arrive at the final multi-modal representations.

Training TriBERT requires the definition of proxy/pretraining tasks and the corresponding losses (see Section 3.2.1). Specifically, while we adopt token masking used in ViLBERT and others, we are unable to define classification targets per token in our visual and pose streams. This is because we only assume per-video labels (*e.g.*, of instruments played) and no access to how those map to attended sounding regions or person instances involved. To address this, we introduce weakly-supervised classification losses for those two streams. Since only one global audio representation is used, this is unnecessary in the audio stream and standard cross-entropy classification can be employed. Finally, motivated by recent works that show that multi-task pretraining is beneficial for ViLBERT [32], we introduce an additional *spectrogram mask prediction* pretraining task which predicts spectrogram masks for each individual audio source from the input spectrogram (bottom block, Figure 1).

**Visual Representations.** Unlike [33], we consider input video frames instead of detected object/bounding box features as our visual input and propose an end-to-end approach to detect and segment objects from each individual frame. Figure 2 illustrates our visual segmentation network which takes in RGB frames as input. To extract global features, we use ResNet50 [20] as the backbone network followed by a $3 \times 3$ *convolution* to generate $H \times W$ visual spatial features which are then fed into the segmentation network. Following [54], we use a decou-

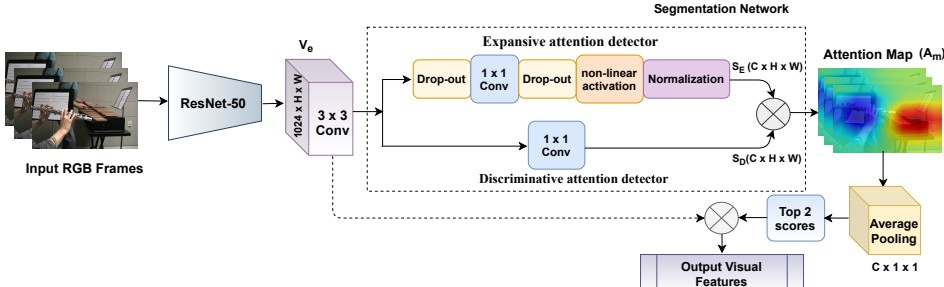

Figure 2: **Visual Segmentation Network.** We consider ResNet50 [20] followed by a $3 \times 3$ *convolution* as our backbone network. It outputs a $H \times W$ spatial visual feature ($V_e$) fed into the segmentation network. Class specific attention map ($A_m$) is generated by the segmentation network and used to pool top two detected object features. We consider the features as visual embeddings for TriBERT.

pled spatial neural attention structure to detect and localize discriminative objects simultaneously. The attention network has two branches: (1) *Expansive attention detector*, which aims to detect object regions and generate the expansive attention map $\mathbf{S}_E \in \mathbb{R}^{C \times H \times W}$ (top branch of Figure 2); and (2) *Discriminative attention detector*, which aims to predict discriminative regions and generate the discriminative attention map $\mathbf{S}_D \in \mathbb{R}^{C \times H \times W}$ (bottom branch of Figure 2). The expansive attention detector contains a drop-out layer followed by a $1 \times 1$ convolution, another drop-out layer, a non-linear activation, and a spatial-normalization layer, defined as follows:

$$\lambda_{(i,j)}^c = F(\mathbf{W}_c^T \mathbf{V}_e(:,i,j) + b^c), \qquad (1) \qquad \alpha_{(i,j)}^c = \frac{\lambda_{(i,j)}^c}{\sum_i^H \sum_j^W \lambda_{(i,j)}^c}, \qquad (2)$$

where $c \in C$ and $F(\cdot)$ denote number of classes and the non-linear activation function, respectively. The final attention map ($\mathbf{A}_m$) is generated as : $\mathbf{A}_m = \mathbf{S}_E \odot \mathbf{S}_D$, where $\odot$ denotes element-wise multiplication. A spatial average pooling is applied on $\mathbf{A}_m$ to generate a classification score for each corresponding class and pooled-out top two class features from spatial-visual feature ($\mathbf{V}_e$). The resultant $3 \times 2 \times 1024$ visual embeddings are used to train our proposed TriBERT architecture, where 3 corresponds to the number of frames and 2 to the number of "objects" per frame.

**Keypoint (pose) Representations.** Our goal is to capture human body and finger movement through keypoint representations. Therefore, we extract 26 keypoints for body joints and 21 keypoints for each hand using the AlphaPose toolbox [12]. As a result, we identify the 2D $(x, y)$ coordinates and corresponding confidence scores of 68 body joints. Following [15], we use Graph CNN to generate semantic context comprising of those joints. Similar to prior work [52] on action recognition, we construct a Spatial-Temporal Graph Convolutional Network $G = \{V, E\}$ where each node $v_i \in \{V\}$ corresponds to the body joint's keypoint and each edge $e_i \in \{E\}$ the natural connectivity between those keypoints. We use 2D coordinates of the detected body joints with confidence scores as input to each node and construct a spatial-temporal graph by: (1) connecting human body joints within a single frame according to body structure; and (2) connecting each joint with the same joint from the consecutive frames. This way, multiple layers of spatial-temporal graph convolutions are constructed to generate higher-level features for human keypoints. We use publicly available code[1] to re-train their model on our dataset and extract body joint features of size $2 \times 256 \times 68$ before the final classification layer (corresponding to two person instances). We apply a linear layer to transform these to $3 \times 2 \times 1024$ input embeddings for pose BERT where 3 corresponds to the number of visual frames and 2 to maximum number of persons per frame.

**Audio Representations.** Consistent with prior works, we use a time-frequency representation of the input audio. We apply STFT [18] to generate the corresponding spectrogram and then transform the magnitudes of the spectrogram into the log-frequency scale for further processing. The size of the final input audio spectrogram is $1 \times 256 \times 256$ and is used in two ways: (1) as an *audio embedding* for audio BERT; and (2) as the *input audio for attention U-net* for the task of sound source separation, which predicts individual audio spectrogram masks (see Figure 1). Before passing to audio BERT, we use a VGGish Network [22] to extract global features for input audio embedding.

---

[1] https://github.com/yysijie/st-gcn

**Tri-modal Co-attention.** Recent works [3, 33] propose co-attentional transformer layers to generate effective representations of vision conditioned on language and vice versa.

In this paper, we introduce a tri-modal co-attentional layer, illustrated on the right, by extending ViLBERT's co-attentional transformer layers [33]. Given intermediate representations for vision, pose, and audio, denoted as $H_V(i)$, $H_P(j)$, and $H_A(k)$, respectively, each stream computes individual query $(Q)$, key $(K)$, and value $(V)$ matrices. The keys and values from two modalities are then concatenated together and fed as input to the multi-head attention block of the third modality. As a result, the block generates attention features conditioned on the other two modalities. We keep the rest of the architecture, such as the feed forward layers, residual connections, *etc.* the same as a standard transformer block, which is then used to generate effective multi-modal features.

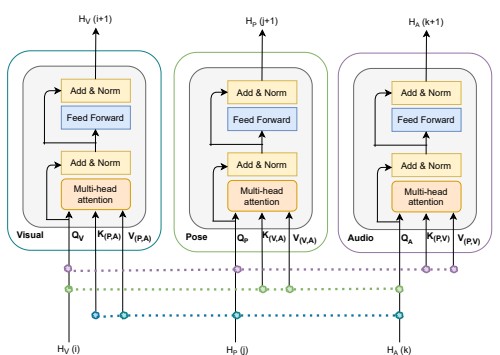

### 3.2.1   Training Tasks

We pre-train TriBERT jointly on two tasks: *instrument classification* and *sound source separation*. Our proposed architecture has three separate streams and each stream performs an individual classification task. To train our TriBERT model, we use the MUSIC21 dataset [56], which contains 21 instruments.

**Weakly-supervised Visual and Pose Classification.** Our visual segmentation network generates attention features for input video frames. We then apply a spatial pooling, and the resulting feature vector is fed into the visual BERT. We use a special <SOS> token at the beginning of the input frame sequence to represent the entire visual input. Following [33], we apply masking to approximately 15% of the input image regions (see Figure 1). The output of the visual BERT is a sequence of hidden representations $h_{v0}, h_{v1}, ..h_{vN}$ conditioned on the pose and audio modalities. We use mean pooling of all hidden representations to perform classification for the detected objects. Similarly, pose BERT generates a sequence of hidden representations $h_{p0}, h_{p1}, ..h_{pN}$ conditioned on the visual and audio modalities, and we apply classification based on the mean pooling of all hidden states. Due to the lack of instance annotations, we cannot use region/pose level supervision. Following [5], we use a weakly-supervised approach to perform region selection and classification.

**Audio Classification.** Since we do not have a sequence of audio embeddings, we artificially create an audio sequence for computational convenience by repeating the VGGish audio feature to generate a sequence of hidden representations $h_{a0}, h_{a1}, ..h_{aN}$ conditioned on the visual and pose modalities. This is done purely for engineering convenience to allow consistent use of tri-modal co-attention across modalities. We then apply audio classification on the mean feature of all hidden representations.

**Multi-modal Sound Source Separation.** We consider sound source separation as one of our initial tasks and follow the "Mix-and-Separate" framework [11, 16, 17, 38, 55], a well-known approach to solve this problem. The goal is to mix multiple audio signals to generate an artificially complex auditory representation and then learn to separate individual sounds from the mixture.

Given two input videos $V_1$ and $V_2$ with accompanying audio $A_1(t)$ and $A_2(t)$, we mix $A_1$ and $A_2$ to generate a complex audio signal mixture $A_m(t) = A_1(t) + A_2(t)$. Suppose $V_1$ has two objects $o_1'$ and $o_1''$ with accompanying audio $a_1'$ and $a_1''$ while $V_2$ has one object $o_2'$ with audio $a_2'$. The goal is to separate sounds $a_1'$, $a_1''$, and $a_2'$ from the mixture $A_m(t)$ by predicting spectrogram masks using attention U-net [37], which takes in the mixed spectrogram as input. Attention U-net contains 7 convolutions and 7 de-convolutions with skip connections. The skip connections use attention gates (AG) comprise simple additive soft attentions to highlight relevant regions of the audio spectrograms. The overhead of attention U-Net over U-Net is fairly minimal. Specifically, in terms of the number of parameters, attention U-Net contains a modest 9% more parameters as compared to U-Net and the inference speed is only 7% slower [37]. The attention U-net outputs the final magnitude of the spectrogram mask (bottom branch in Figure 1) guided by audio-visual-pose features. Following [15], we adopt a self-attention based early fusion between the bottle-neck of attention U-net with the fused features (*i.e.* concatenation of features) corresponding to the <SOS> tokens of three BERT streams.

We combine the predicted magnitude of the spectrogram mask from attention U-net with the phase of the input spectrogram and then use inverse STFT [18] to get back the wave-form of the prediction.

**Training Objective.** We consider weakly-supervised classification for the visual and pose modalities. Following [5], we use two data streams from the hidden state of each modality. The first stream corresponds to a class score ($\beta_{class}$) for each individual region to perform *recognition*. This is achieved by a *linear layer* followed by a *softmax* operation (see Eq. 3). The second stream computes a probability distribution ($\beta_{det}$) for performing a proxy *detection*. This is done by using another *linear layer* followed by another *softmax* operation (see Eq.4) as follows:

$$\beta_{class}(\mathbf{h}^c)_{ij} = \frac{e^{h^c_{ij}}}{\sum_{t=1}^{C} e^{h^c_{tj}}}, \qquad (3) \qquad \beta_{det}(\mathbf{h}^d)_{ij} = \frac{e^{h^d_{ij}}}{\sum_{t=1}^{|R|} e^{h^d_{tj}}}, \qquad (4)$$

where $h^c \in \mathbb{R}^{C \times |R|}$, $h^d \in \mathbb{R}^{C \times |R|}$ and $C$ denotes the number of classes. We then aggregate the recognition and detection scores to predict the class of all image regions as follows: $\beta^R = \beta_{class}(\mathbf{h}^c) \odot \beta_{det}(\mathbf{h}^d)$, where $\odot$ denotes an element-wise product of the two scoring metrics. Finally we apply *BCE-loss* [10] to train visual and pose BERT. For audio classification, we consider a classification layer to predict audio classes and similarly apply *BCE-loss* to train audio BERT.

For the sound separation task, our goal is to learn separate spectrogram masks for each individual object. Following [55], we use a binary mask which effectively corresponds to hard attention and use per-pixel sigmoid cross entropy loss (*BCE-loss*) to train the network.

**Implementation Details.** We used PyTorch to implement our network[2]. We consider three[3] random consecutive frames with size $224 \times 224 \times 3$ as our input sequence for visual and pose BERT and use pre-trained ResNet50 [20] to extract global visual features for further processing. For the pose stream, we first predict 2D coordinates of body and finger key points of each frame using AlphaPose [12] and then use graph CNN [52] to generate feature vectors for each keypoint. Similar to prior works [15, 55], we sub-sample audio signals to 11KHz to reduce the computational cost and then select approximately *6s* of audio by random cropping. To follow the "Mix-and-Separate" framework [11, 16, 17, 38, 55], we mix audio inputs and generate a time-frequency audio spectrogram using STFT with a Hann window size of 1022 and a hop length of 256. We then transform the spectrogram into the log-frequency scale to obtain the final $256 \times 256$ time-frequency representation. The transformers for visual/pose and audio have a hidden state size of 1024 and 512, respectively, with 8 attention heads. We use the Adam optimizer with an initial learning rate of $1e^{-5}$ and batch size of 12 to train the network on 4 GTX 1080 GPUs for $6k$ epochs. Training takes approximately 192 hours.

### 3.2.2 Runtime Inference

We use the MUSIC21 dataset [56] to train our network on two pretraining tasks: classification and sound source separation. We can use this network directly for sound separation on MUSIC21. We also fine-tune the pre-trained TriBERT on the MUSIC dataset [55] with 11 audio classes, which is a sub-set of the MUSIC21 dataset. We follow a fine-tuning strategy where we modify the classification layer from each pre-trained stream and then train our proposed model end-to-end with a learning rate of $1e^{-7}$ for 1500 epochs while keeping the rest of the hyper-parameters the same as the initial task.

## 4 Experiments

**Datasets.** We consider the MUSIC21 dataset [56], which contains 1365 untrimmed videos of musical solos and duets from 21 instrument classes for the initial training of our TriBERT architecture. For fine-tuning, we use the MUSIC dataset [55], which is a subset of MUSIC21, containing 685 untrimmed videos of musical solos and duets from 11 instrument classes.

---

[2]https://github.com/ubc-vision/TriBERT

[3]BERT-based architectures, including ours, require large GPU memory and longer training time. Therefore, we use only three frames to reduce computational cost, but the number of frames can be easily increased with the same architecture (if resources allow). Further, we would like to highlight that a pose feature for one frame, actually takes into account T=256 frames of poses using a Spatial-Temporal Graph Convolutional Network. Therefore long-term contextual pose information is taken into account [52].

Table 1: **Sound separation results on the MUSIC21 test set.** SDR / SIR / SAR are used to report performance. Separation accuracy is captured by SDR and SIR; SAR only captures the absence of artifacts.

| Methods | Single-Source | | | Multi-Source | | |
|---|---|---|---|---|---|---|
| | SDR (↑) | SIR (↑) | SAR (↑) | SDR (↑) | SIR (↑) | SAR (↑) |
| Sound-of-Pixels [55] | 6.57 | 12.82 | 10.78 | 5.73 | 12.11 | 10.10 |
| MUSIC-Gesture [15] | 8.08 | 15.27 | 11.29 | 6.72 | 14.03 | 9.68 |
| Ours | **10.09** | **17.45** | **12.80** | **7.66** | **14.54** | **11.06** |

Table 2: **Sound separation results on the MUSIC test set.** We use TriBERT with pre-trained weights from the MUSIC21 dataset and then fine-tune this model using the MUSIC train set.

| Methods | SDR (↑) | SIR (↑) | SAR (↑) |
|---|---|---|---|
| NMF-MFCC [45] | 0.92 | 5.68 | 6.84 |
| AV-Mix-and-Separate [16] | 3.23 | 7.01 | 9.14 |
| Sound-of-Pixels [55] | 7.26 | 12.25 | 11.11 |
| CO-SEPARATION [16] | 7.64 | 13.8 | 11.3 |
| Mask Co-efficient with seg net [42] | 9.29 | 15.09 | 12.43 |
| Ours (after fine-tune) | **12.34** | **18.76** | **14.37** |

## 4.1 Experiments for Sound Separation

**Evaluation Metrics.** We use three common metrics to quantify the performance of sound separation: Signal-to-Distortion Ratio (SDR), Signal-to-Interference Ratio (SIR), and Signal-to-Artifact Ratio (SAR). We report all of the results with the widely used mir_eval library [41].

**Baselines.** The MUSIC21 dataset contains 1365 untrimmed videos, but we found 314 of those to be missing. Moreover, the train/val/test split was unavailable. As a result, for fair comparison, we trained our baselines [15, 55] with the available videos using an 80/20 train/test split. We use publicly available code[4] to train "Sound-of-Pixels" [55]. For "MUSIC-Gesture" [15], we re-implemented the model by extracting pose features using Graph CNN [52]. Our reproduced results are comparable with those reported[5]. For the MUSIC dataset, we follow the experimental protocol from [42] and consider their reported results as our baselines.

**Quantitative and Qualitative Results.** Table 1 shows the quantitative results for the sound separation pre-training task on the MUSIC21 dataset. Here, we include the performance of our method and baselines when we use only single-source videos (solos) or multi-source (solos+duets) to train all models. Our TriBERT outperforms (10.09 vs 8.08 for single-source in SDR) baseline models in all evaluation metrics. We then fine-tune our model on the MUSIC dataset with a train/val/test split from [16] (see Table 2). Our model again outperforms all baselines in all metrics (12.34 vs 9.29 in SDR). Figure 3 illustrates the corresponding qualitative results. The 1st, 2nd, and 3rd columns show the mixed video pairs and accompanying audio mixture, respectively. Columns 4 and 5 illustrate the ground-truth spectrogram mask while columns 6/7 and 8/9 show the predicted spectrogram mask by [15] and our method, respectively. Finally, the ground truth spectrogram, predicted spectrogram by [15], and our method are illustrated in columns 10/11, 12/13, and 14/15, respectively. It is clear that TriBERT, both quantitatively and qualitatively, outperforms the state-of-the-art in sound separation.

## 4.2 Multi-modal Retrieval

**Retrieval Variants.** In this experiment, we analyze the semantic alignment between the 3 modalities that TriBERT learns to encode. This is done through cross-modal retrieval, where given a single or a pair of modality embeddings, we attempt to identify the matching embedding from a different modality. We consider 5 variants: audio → vision, vision → audio, audio → pose, pose → audio, and vision+audio → pose. Throughout this section, we refer to the embedding we have as the *query*

---

[4]`https://github.com/hangzhaomit/Sound-of-Pixels`

[5]The reported SIR score in [15] is 15.81, which is close to our reimplementation of their method which achieves a score of 15.27. Our reproduced SDR score is a bit lower, compared to the 10.12 reported in [15]. However, this is perhaps expected given that 23% of the dataset was missing.

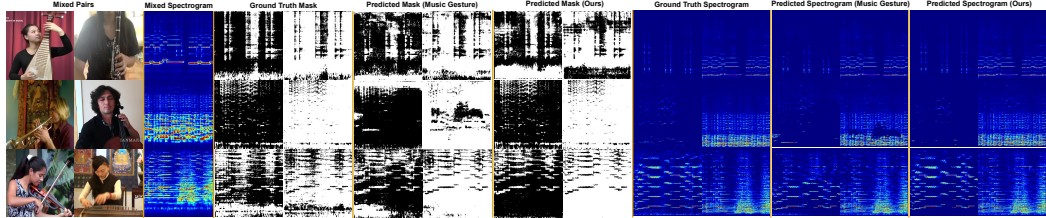

Figure 3: **Qualitative sound separation results on the MUSIC21 test set.** Here, we show a comparison between our results and Music-Gesture [15]. See text for details.

embedding and the embedding we want to retrieve as the *result* embedding. We train and evaluate on the MUSIC21 dataset, using the same 80-20 train-test split used to learn TriBERT.

We consider 2 types of embeddings for the 3 modalities. First, we use the transformer-based embeddings, consisting of the concatenations of the hidden representations $h_{v0...v3}$, $h_{p0...p3}$, and $h_{a0...a3}$ for visual, pose, and audio, respectively. Additionally, we establish a baseline by training with the embeddings used as input to the three BERT streams. This baseline can be viewed as an ablation study for the transformer layers.

**Retrieval Training.** Similar to [33], we train using an $n$-way multiple-choice setting. Here, $n$ depends on the variant of the retrieval task, where $n = 4$ for the vision+audio to pose variant and $n = 3$ for the four remaining single-modality variants. In either case, one positive pair is used and $n - 1$ distractors are sampled. Further details are provided in the Supplemental Materials. We use an MLP that takes as input a fusion representation of both the query and result embeddings, computed as the element-wise product of the two. The module then outputs a single logit, interpreted as a binary prediction for whether the query and result embeddings are aligned. For the vision+audio $\rightarrow$ pose variant, an additional MLP, based on [15], is used to combine the vision and audio embeddings before the final element-wise product with the pose embedding. Additionally, since both the transformer-based and pre-transformer embeddings are not consistent in shape across the three modalities, we also use linear layers as required to transform them to a consistent one. This overall retrieval network is trained end-to-end. For each multiple choice, the network computes an alignment score, after which a softmax is applied across all $n$ scores. We train using a cross-entropy loss for 750 epochs with a batch size of 64 using the Adam optimizer with an initial learning rate of 2e-5.

**Retrieval Results.** Figure 4 shows the qualitative results for two variants of retrieval. Additionally, Table 3 shows quantitative results for the 5 retrieval variants using the transformer-based representation, the baseline pre-transformer representation, and also a model that simply selects randomly from the pool. We see that retrieval using the transformer-based embeddings results in significantly better performance than the pre-transformer ones. This shows that the tri-modal co-attention modules are an integral component in learning a semantically meaningful relationship between the three modalities.

Notably, in Table 3, we can see that vision+audio $\rightarrow$ pose is worse than audio $\rightarrow$ pose in top-1 accuracy. The performance of the two models is not necessarily directly comparable. Specifically, there are two issues that should be considered:

- The input dimensionality and number of parameters of the vision+audio retrieval model is significantly larger, with an additional MLP layer used for fusion. This means that the vision+audio model is more prone to over-fitting, exhibited in the lower performance for top-1. Note that the top-5 and top-10 performance of vision+audio $\rightarrow$ pose is better.
- The number of distractors $(n - 1)$ is different in the two settings. For single-modality retrieval variants, we use two distractors (negative pairings); while for the two-modality variant, we use three distractors. This may also marginally affect the performance, since in the $n$-way classification, having more distractors puts more focus on the negatives.

However, we want to stress that the goal of these experiments is not to compare which modality or combination of modalities are best for retrieval. Instead, the goal is to illustrate the effectiveness of the TriBERT representations. Each of the five retrieval models is simply an instance of a retrieval task. We can use any alternative (more sophisticated) models for retrieval here. The key observation is that in all five cases, TriBERT representations perform significantly better in retrieval compared with baseline representations (used as input to TriBERT). This is strong evidence that TriBERT representations are

Table 3: **Multi-modal retrieval on the MUSIC21 test set.** Top-$k$ accuracy results for 5 retrieval variants. For each variant, retrieval on both the transformer-based (bolded) and pre-transformer (unbolded) embeddings were evaluated. Also shown is the accuracy of a random selection model.

| Retrieval Variant | Top-1 Accuracy (random = 0.48) | | Top-5 Accuracy (random = 2.38) | | Top-10 Accuracy (random = 4.76) | |
|---|---|---|---|---|---|---|
| Audio → Vision | **68.10** | 1.43 | **93.81** | 10.48 | **98.57** | 17.62 |
| Vision → Audio | **59.52** | 4.76 | **89.52** | 18.57 | **92.38** | 31.90 |
| Audio → Pose | **63.81** | 2.38 | **86.67** | 6.67 | **94.29** | 10.95 |
| Pose → Audio | **54.29** | 1.90 | **85.24** | 9.05 | **86.67** | 14.29 |
| Vision + Audio → Pose | **54.29** | 3.33 | **90.00** | 9.05 | **96.19** | 15.24 |

effective. We make no claims with regards to optimality of the retrieval formulation or objective; it is simply used as a proxy for evaluating TriBERT representations.

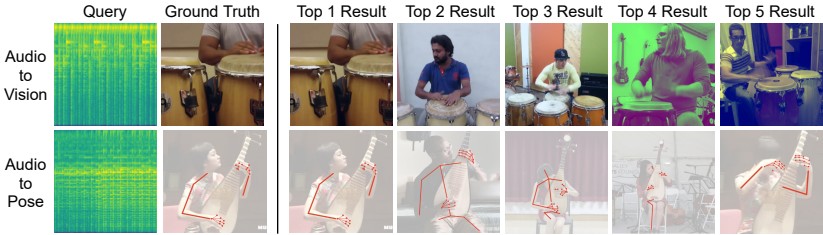

Figure 4: **Qualitative cross-modal retrieval results on the MUSIC21 test set.** The similarities (same instrument class) between results in the top-5 retrieval pool show that our transformer-based representations have learned a semantically meaningful relationship between the modalities.

## 5 Conclusion

In this paper, we introduce TriBERT, a three-stream model with tri-modal co-attention blocks to generate a generic representation for multiple audio-visual tasks. We pre-train our model on the MUSIC21 dataset and show that our model exceeds state-of-the-art for sound separation[6]. We also find that TriBERT learns more generic and aligned multi-modal representations, exceeding on the cross-modal audio-visual-pose retrieval task. In this work, we limit ourselves to two datasets and fundamental audio-visual tasks. In the future, we plan to consider using more datasets and expanding to a broader set of tasks (*e.g.*, generation). The role of positional embeddings should also be explored.

**Acknowledgments:** This work was funded in part by the Vector Institute for AI, Canada CIFAR AI Chair, NSERC Canada Research Chair (CRC) and an NSERC Discovery and Discovery Accelerator Supplement Grants.

Resources used in preparing this research were provided, in part, by the Province of Ontario, the Government of Canada through CIFAR, and companies sponsoring the Vector Institute www.vectorinstitute.ai/#partners. Additional hardware support was provided by John R. Evans Leaders Fund CFI grant and Compute Canada under the Resource Allocation Competition award.

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
