# TriBERT: Full-body Human-centric Audio-visual Representation Learning for Visual Sound Separation (Supplementary Materials)

**Tanzila Rahman**[1,3]     **Mengyu Yang**[2,3]     **Leonid Sigal**[1,3,4]
[1]University of British Columbia     [2]University of Toronto
[3]Vector Institute for AI     [4]Canada CIFAR AI Chair
trahman8@cs.ubc.ca, my.yang@mail.utoronto.ca, lsigal@cs.ubc.ca

## 1   Negatives Sampling Details on Cross-Modal Retrieval

Recall that for the $n$-way multiple choice setting, $n - 1$ choices are negative pairs and only one pair is positive. Accordingly, for $n = 4$, 3 distractors are sampled, each with an incorrect pose embedding, while the 4th choice contains the matching pose embedding for the given vision and audio embeddings. In other words, the fusion embedding consisting of the vision and audio embeddings is kept as the anchor while negatives are sampled from the pose embeddings only. Of the 3 negative pose embeddings, 2 are considered "easy" negatives, sampled randomly from the entire training set, while the last one is a "hard" negative, sampled randomly from a pool of 25 embeddings corresponding to the 25 nearest neighbours of the anchor vision embedding. In the $n = 3$ case, 2 hard negatives and no easy negatives are used, with the same nearest neighbour sampling method based on the anchor embedding. Figure 1 shows a diagram of the training scheme for the cross-modal retrieval module.

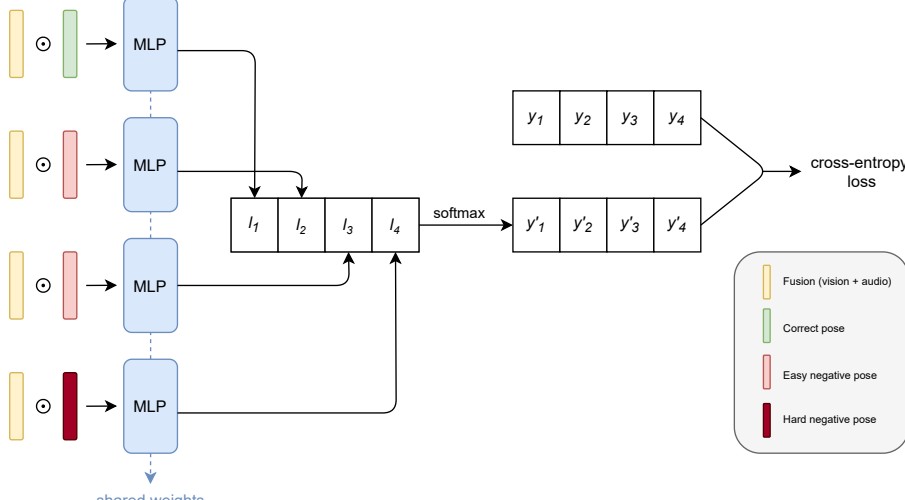

Figure 1: **Overview of the training scheme for the cross-modal retrieval module in the $n = 4$ case.** Each multiple choice consists of the correct vision+audio fusion embedding along with a pose embedding. The 4 choices: 1 containing the correct pair, 2 containing easy negatives, and 1 containing a hard negative is passed as input into a MLP network. The network outputs a score for each pair, after which a softmax is applied across all 4 scores. We train this network in a siamese fashion (*i.e.*, each MLP block in the figure shares the same weights) using cross entropy loss. In our experiments, we use a 4-layer MLP with Tanh activation.

35th Conference on Neural Information Processing Systems (NeurIPS 2021).

## 2 Additional Experiments on TriBERT

### 2.1 Ablation to Justify Tri-modal Design.

To justify the tri-modality design, we have conducted additional experiments by masking either the visual or pose modality and only using the remaining two modalities (*i.e.*, audio/visual or audio/pose) for sound separation during fine-tuning on the MUSIC dataset (see Table 1). Masking is equivalent to erasing a modality.

Table 1: **Experimental results if one of the modality is erased.** Here we use multi-source videos (solos + duets) to fine-tune on MUSIC dataset.

| Type of Masking | SDR ($\uparrow$) | SIR ($\uparrow$) | SAR ($\uparrow$) |
|---|---|---|---|
| Masking is used for visual modality | 7.82 | 14.39 | 10.65 |
| Masking is used for pose modality | 12.06 | 18.34 | 14.17 |
| 15% random masking for both visual and pose modality | **12.34** | **18.76** | **14.37** |

In this paper, we are using sound separation as our primary task. Therefore, we do not consider masking for the audio modality. We do 15% random masking for both visual and pose modality by default as part of our TriBERT model training. Overall, one can see that visual modality is contributing more, comparatively, than the pose modality, but both modalities are useful in improving the performance. When pose modality is used alone (top row in Table 1) the performance is still reasonable. This suggests some level of redundancy in pose and visual information, rather than lack of information in the pose modality.

### 2.2 Experiments with Different Fusion.

To generate audio-visual-pose "Fused Features" (see Figure 1 in main paper), we considered two types of simple fusions: (1) concatenation; and (2) multiplication. For each fusion, we pool-out *<SOS>* token features from visual, pose and audio BERT; and apply multiplication or concatenation on these features (see Table 2). We find that concatenation is performing slightly better than multiplication. Therefore, in this paper we use simple concatenation to generate audio-visual-pose features and fused with bottleneck features of attention U-Net [2] for further processing.

Table 2: **Experimental results for different fusion.** Here we use multi-source videos (solos + duets) to train the models.

| Fusion Methods | SDR ($\uparrow$) | SIR ($\uparrow$) | SAR ($\uparrow$) |
|---|---|---|---|
| Multiplication | 7.56 | **14.61** | 10.78 |
| Concatenation | **7.66** | 14.54 | **11.06** |

### 2.3 Differences Between MUSIC21 and MUSIC Dataset.

Even though the MUSIC dataset is a subset of MUSIC21, this subset has notably different statistical characteristics (different number of classes, relative distribution of data, *etc.*). As a result of this, a model trained on MUSIC21 is not necessarily going to produce optimal performance on the MUSIC dataset. To illustrate this we report sound separation performance on MUSIC dataset before and after fine-tuning in Table 3.

Table 3: **Experimental results on MUSIC dataset before and after fine-tuning.** Even though given that it is a subset of MUSIC21.

| | SDR ($\uparrow$) | SIR ($\uparrow$) | SAR ($\uparrow$) |
|---|---|---|---|
| Before fine-tuning | 11.81 | 17.53 | 14.69 |
| After fine-tuning | **12.34** | **18.49** | **14.63** |

We can see that after fine-tuning, sound separation results have improved in terms of SDR and SIR. SAR is responsible for capturing absence of artifacts and it can be higher even though separation results are poor. Note that if MUSIC21 and MUSIC had similar statistics and the model trained on MUSIC21 was "sufficient", the performance with fine-tuning would not improve. This is not the case. Therefore MUSIC dataset can be considered to be a "down-the-stream" dataset on which effectiveness of features, using fine-tuning, can be tested.

## 2.4 Effectiveness of Learned Generic Representations for Audio-visual Classification.

We conducted additional experiments to show the effectiveness of learned generic representations. We extract learned TriBERT representations (*i.e.*, audio/visual/pose features) from the MUSIC21 dataset and use a simple MLP for instrument classification. We consider raw uni-modal features (which are input to TriBERT) as the baselines. We then compare these baselines to both uni-modal masked and multi-modal TriBERT features (see Table 4). In doing so, no fine-tuning is employed (this can further improve results).

Table 4: **Experimental results for audio-visual classification.** Here we consider raw uni-modal features without BERT stream as our baseline and compare with TriBERT features.

| Features | Top-1 | Top-5 | Top-10 |
|---|---|---|---|
| Audio feature w/o audio BERT stream | 56.87 | 84.36 | 93.84 |
| TriBERT audio feature with visual/pose masked | 49.29 | 84.31 | 92.42 |
| TriBERT audio feature (our model) | **86.26** | **94.79** | **98.10** |
| Vision feature w/o visual BERT stream | 41.23 | 63.03 | 77.25 |
| TriBERT visual feature with audio/pose masked | 81.52 | 92.89 | 95.26 |
| TriBERT visual feature (our model) | **87.20** | **94.31** | **96.68** |
| Pose feature w/o pose BERT stream | 39.81 | 76.78 | 91.47 |
| TriBERT pose feature with audio/visual masked | 35.07 | 68.25 | 84.36 |
| TriBERT pose feature (our model) | **87.68** | **96.21** | **98.58** |

We can make a couple of observations here. First, audio and pose modalities appear to not benefit from within-modality transformer interactions of features. This makes sense since effectively we only have a single unique audio feature and pose features are highly redundant due to their long temporal context. The slightly lower performance over the "raw" input features is simply due to the learned co-attention across modalities (since in this experiment we do not fine-tune masked model variants). Notably, the visual features do tend to benefit from within-modality interactions among frames, because of the complementary and instantaneous nature of visual representation.

Most importantly, one can see very significant improvements in classification when TriBERT is able to co-attend across modalities (our model). This is a strong indication of the benefits of multi-modal features produced by our TriBERT architecture. The improved performance and additional discriminative ability here can be attributed directly to cross-modal co-attention learned and induced by TriBERT.

## 2.5 Additional Qualitative Results.

We visualize the segmentation localization qualitatively in Figure 2. To the best of our knowledge, prior SOTA approaches did not provide quantitative results for the sound source localization. In fact, MUSIC21 and MUSIC datasets do not have annotations for the sounding objects. As such, we are not able to provide quantitative comparison for object detection.

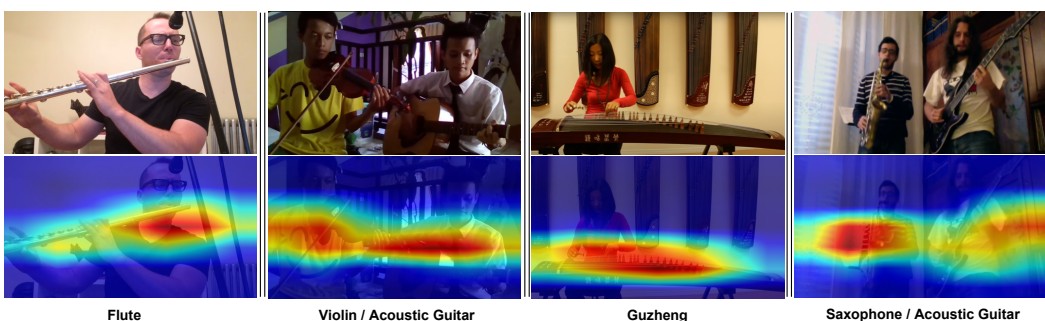

Figure 2: **Attention object map.** Attention map (red higher attention) with class labels (in bottom) from our learned visual segmentation network which focus on objects that make sounds.

We would also like to show some more qualitative results in addition to our main paper in Figure 3 and 4. We also compare our results with Music Gesture [1] (*i.e.*, most recent baseline). In both

figures, the first and second rows show input mixed video pairs with accompanying audio respectively. The 3rd, 4th and 5th rows present ground-truth mask and predicted spectrogram mask generated by ours method and Music Gesture [1] respectively. Finally, ground truth and predicted spectrogram using our method and Music Gesture [1] are shown in the 6th, 7th and 8th rows accordingly. One can clearly see that predicted mask and spectrogram generated by our method are closer to ground truth.

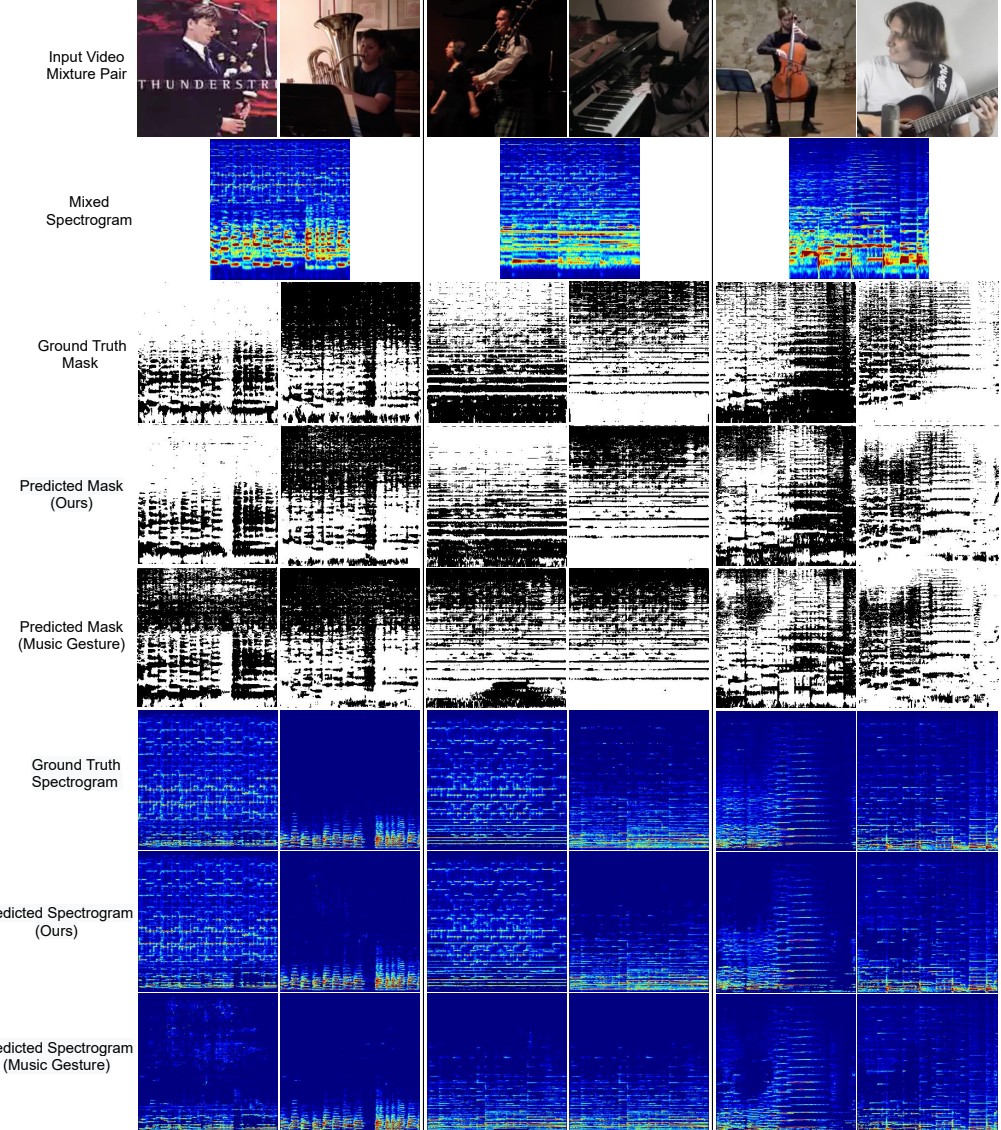

Figure 3: **Qualitative sound separation results on the MUSIC21 test set.** Here, we show a comparison between our results and Music-Gesture [1].

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

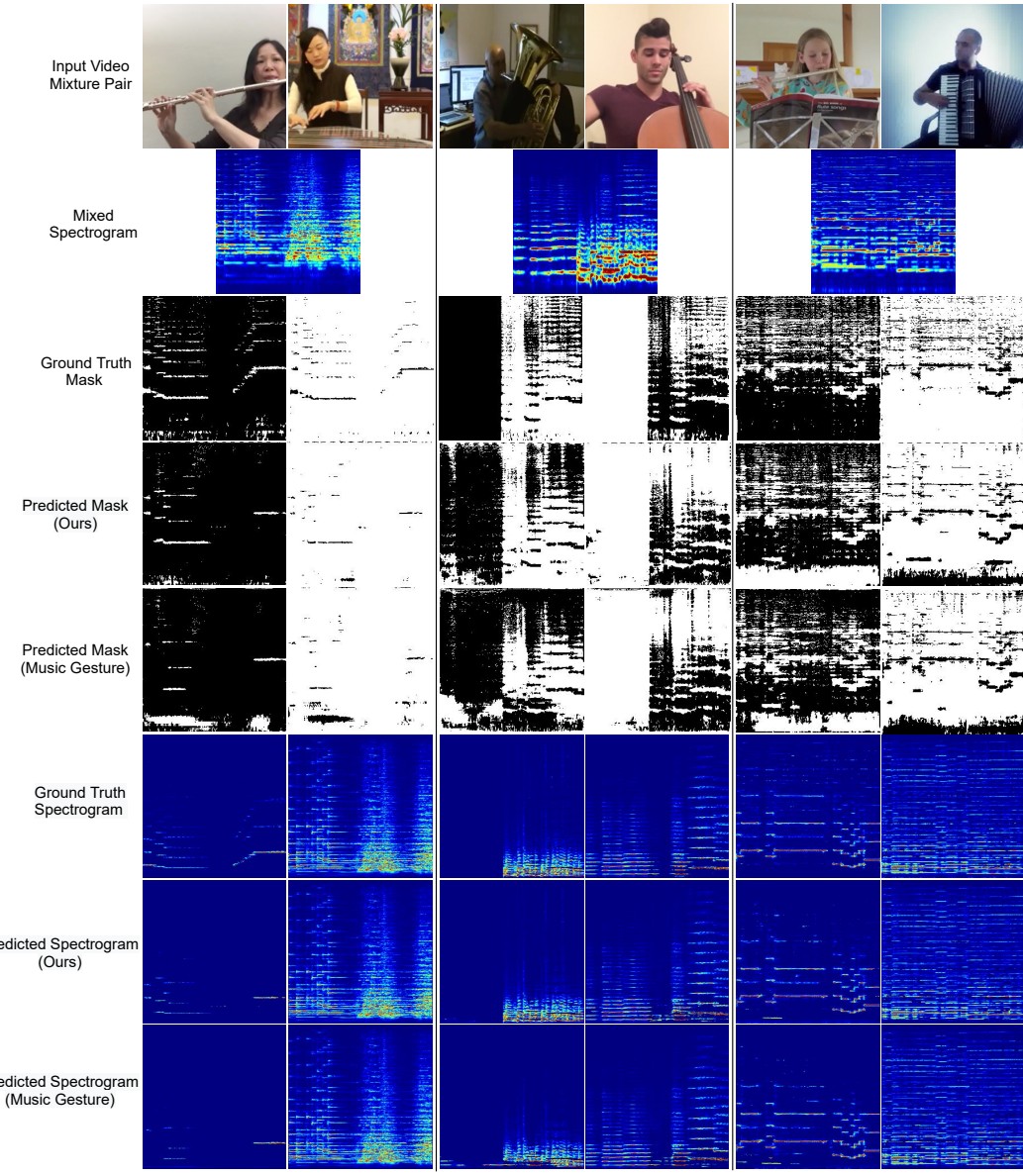

Figure 4: **Some additional sound separation results on the MUSIC21 test set.** Again, we compare our results with Music-Gesture [1].