# OpenReview forum: "TriBERT: Human-centric Audio-visual Representation Learning"
_NeurIPS.cc/2021/Conference — NeurIPS 2021 Poster_

### Official Review · Reviewer_voNw · 2021-07-16

**Rating:** 6
**Confidence:** 3

**Summary:**

The paper presents TriBERT a BERT-like model for learning multi-model representations of three modalities simultaneously, namely: vision, pose and audio. The architecture of TriBERT takes inspiration from the visual-linguistic model VilBERT (notably known for the applications in VQA) in terms of its Transformer architecture and of its multi-modal attention (the latter is extended in the paper to 3 modalities instead of 2).

TriBERT training comprises four losses to optimize. In particular, after the projection of the 3 unimodal feature sequences into the joint feature space by the multimodal transformer, there are 3 sequences of features, namely: <h_v>, <h_p> and <h_a> for respectively enriched visual, pose and audio features. Then, the <h_v> and <h_p> features are (independently) trained to localize and predict the sound source (the musical instrument(s) which is (are) played in the recording) using 2 (independent) BCE losses. It should be noted that the localization part is learnt in a weakly-supervised manner as only global instrument classification annotations are available in the training data. The <h_a> audio features are trained to classify the sound source (the 3rd BCE loss).

Finally, the 4th supervision task is the multi-modal sound source separation one. The original audio spectrogram is mixed with other audio spectrogram(s) and the task consists in predicting a spectrogram mask which would separate the original audio from the others (the 4th BCE loss). Following [37], for this last task, a U-Net neural architecture is used. U-Net takes the mixed audio spectrogram at its input and the fusion of <h_v>, <h_p> and <h_a> at its bottleneck representation to predict the desired spectrogram mask.

Once trained, TriBERT is evaluated on the MUSIC21 dataset for the sound separation task, and it also fine-tuned and evaluated for the multi-modal retrieval downstream task.


**Limitations And Societal Impact:**

I do not see any negative social impact for this research.

**Main Review:**

*Originality*

The paper has a primary original contribution: in particular, to the best of my knowledge, it is the 1st study to report a conclusive approach for training of a tri-modal BERT-like model on visual, audio and pose modalities with impressive results on such a complicate downstream task as multi-modal retrieval.

The paper has also two other minor contributions. Firstly, it demonstrates that unimodal visual features can be learnt from scratch in an end-2-end manner (while a common approach is to extract object features with a pretrained object detector such as Faster-RCNN as it is done in VilBERT, for example), and secondly, they illustrate that sound source prediction, separation and localization are powerful supervision tasks which are enough to train a complex Transformer architecture with relatively little annotations (only the tags of played instruments).

*Quality*

I believe that the authors have properly cited and compared their approach to all significant related work, most notable to [15] and [55]. Moreover, the approach of [15] has been re-implemented and retrained in order to use the same unimodal features for the pose modality (the ones learnt with a graph CNN on keypoints extracted with the AlphaPose toolbox).

All in all, in my opinion, the paper is technically sound.

*Clarity*

The paper is very well written and easy to follow. The authors commit to publishing the source code upon the paper acceptance.

*Typos*

1.	Line 93: “of of” - > “of”.

*Significance*

To my knowledge, this is the 1st paper which has successfully managed to learn joint representations of vision, pose and audio inputs with a single BERT-like architecture. Moreover, only very shallow annotations (the musical instrument type) of the training dataset (MUSIC21) have been used for training. Given the convincing experimental results on the sound separation task where the strong Sound-of-Pixels [55] and MUSIC-Gesture [15] baselines have been largely outperformed, as well as the multi-modal retrieval results reported in Table 3 which demonstrate that TriBERT managed to learn a joint feature space for the three modalities, I believe that this paper will be useful for the community as a 1st example of what might become a new standard in audio-visual processing (as it was the case for VilBERT in the visual-linguistic domain). Therefore, I am rather inclined to recommend the acceptance of this paper.

At the same time, I would like to highlight the following limitations and suggestions / questions for the authors:

1.	The experiments are done only on a single dataset (MUSIC is actually a subset of MUSIC21). The results would have been even more convincing, had there been another dataset for evaluation.

2.	I would be interested to see an ablation on the contributions of each modality to the final sound separation results reported in Tables 1 and 2. Thus, I believe that the pose and acoustic modalities are much more important than the visual one for this task. Does the visual modality really contribute in the final score?

3.	It would also be informative to see some qualitative results on the sound localization given the fact that this task has been learnt in a weakly supervised manner (without explicit annotations on the localization).

4.	Finally, how can you explain that the retrieval Top-1 accuracy for the pose modality given only audio modality is higher than given both audio and visual modalities (Table 3)?



**Time Spent Reviewing:**

6

---

> ### Author Response · Authors · 2021-08-10
> **Response to Reviewer voNw**
>
> We would like to thank the reviewer for the valuable feedback. We address individual concerns below.
>
> * 1a. The experiments are done only on a single dataset (MUSIC is actually a subset of MUSIC21)
>
> Even though the MUSIC dataset is a subset of MUSIC21, this subset has notably different statistical characteristics (different number of classes, relative distribution of data, etc.). As a result of this, a model trained on MUSIC21 is not necessarily going to produce optimal performance on the MUSIC dataset. To illustrate this we report sound separation performance on MUSIC dataset before and after fine-tuning:
>
> ---
>
> Before fine-tuning      &emsp;&emsp;&emsp;&emsp;&emsp;         SDR: 11.81       &emsp;&emsp;&emsp;            SIR: 17.53               &emsp;&emsp;&emsp;  SAR: 14.69
>
> ---
>
> After fine-tuning         &ensp;&emsp;&emsp;&emsp;&emsp; &emsp;        SDR: 12.34        &emsp;&emsp;&emsp;           SIR: 18.49     &emsp;&emsp;&emsp;            SAR: 14.63
>
> ---
>
> We can see that after fine-tuning, sound separation results have improved in terms of SDR and SIR.  SAR is responsible for capturing absence of artifacts and it can be higher even though separation results are poor. Note that if MUSIC21 and MUSIC had similar statistics and the model trained on MUSIC21 was “sufficient”, the performance with fine-tuning would not improve. This is not the case. Therefore MUSIC dataset can be considered to be a “down-the-stream” dataset on which effectiveness of features, using fine-tuning, can be tested.
>
>
> * 1b. Use another dataset for evaluation
>
> We were unable to run experiments on another dataset (e.g. UCF101 or AudioSet) due to the time limit during the rebuttal, we have conducted additional experiments on instrument classification on the MUSIC21 dataset which has been discussed in response to Reviewer 8Z91, Question 2.
>
>
> * 2. Ablation on the contribution of each modality to the final sound separation
>
> We have conducted additional experiments by masking either the visual or pose modality and only using the remaining two modalities (i.e., audio/visual or audio/pose) for sound separation. The results are as follows:
>
> ---
>
> SDR: 7.82  &ensp;&emsp;&emsp;&emsp; SIR: 14.39  &emsp;&emsp;&emsp; SAR: 10.65 &emsp;&emsp;&emsp;(Masking is used for visual modality)
>
> SDR: 12.06 &emsp;&emsp;&emsp; SIR: 18.34 &emsp;&emsp;&emsp; SAR: 14.17 &emsp;&emsp;&emsp;(Masking is used for pose modality)
>
> SDR: 12.34 &emsp;&emsp;&emsp; SIR: 18.76 &emsp;&emsp;&emsp; SAR: 14.37 &emsp;&emsp;&emsp;(15% random masking for both visual and pose modality)
>
> ---
>
> Note, we do 15% random masking for both visual and pose modality by default as part of our original TriBERT model training. One can see that visual modality is contributing more, comparatively, than the pose modality, but both modalities are useful in improving the sound separation performance.
>
>
> * 3. Qualitative results on the sound localization
>
> We apologize for not including these results in the supplemental. Due to the nature of OpenReview, we could not upload these results as part of the rebuttal without including external links (which would violate the blind nature of the review process). Upon acceptance of the paper, we will add these to the Supplemental Material.
>
>
> * 4. How can you explain that the retrieval Top-1 accuracy for the pose modality given only audio modality is higher than given both audio and visual modalities (Table 3)
>
> The performance of the two models is not necessarily directly comparable. Specifically, there are two issues that should be considered:
>
> - The input dimensionality and number of parameters of the V+A retrieval model is significantly larger, with an additional MLP layer used for fusion (see Line 354). This means that V+A model is more prone to overfitting. This is exhibited in slightly  lower performance for top1; notably top5 and top10 performance of V+A -> P is better.
>
> - The number of hard-negatives (n) is different in the two settings. For single-modality retrieval variants we use two distractors (negative pairings); while for two-modality variant we use three destractors. This may also marginally affect the performance, since in the n-way classification, having more distractors puts more focus on the negatives.
>
> However, we want to stress that the goal of these experiments isn’t to compare which modality or combination of modalities are best for retrieval. Instead, the goal is to illustrate the effectiveness of the TriBERT features. Each of the five retrieval models is simply an instance of a retrieval task. We can use any alternative (more sophisticated) models for retrieval here. The key observation is that in all five cases TriBERT features are SIGNIFICANTLY better in retrieval as compared with baseline features (used as input to TriBERT). This is strong evidence that TriBERT feature representations are effective. We make no claims with regards to optimality of the retrieval formulation or objective; it is simply used as a proxy for evaluating TriBERT features.
>
> * Typos: We will fix this in the revision.

---

### Official Review · Reviewer_8Z91 · 2021-07-17

**Rating:** 6
**Confidence:** 3

**Summary:**

The goal of this work is to learn human-centric audio-visual representations. The authors claim to be first in using transformer models in the context of granular audio-visual detection or segmentation tasks (e.g. sound source separation). The authors propose TriBERT, which is a three-stream model with tri-modal attention to generate generic representaitons for AV tasks. The three modalities are audio, video and pose (keypoints). The model is trained with weak supervision (video level labels). The model gives state-of-the-art results on sound source separation.

**Ethical Concerns:**

No concern

**Limitations And Societal Impact:**

Addressed in Section 5

**Main Review:**

The paper introduces BERT-like contextualized representation learning for the audio-visual domain. This is first in the context of AV source separation to my knowledge. The method appears to be effective for the task.

The architecture is based on existing work (VilBERT), but adds additional components such as segmentation network, spectrogram prediction mask and pose input.

The results are strong on the datasets shown. However, as the authors noted, the experiments are limited to only two datasets, one of which cannot be reproduced completely due to missing files (line 307). Ablations are also not given to justify design choices.

The results would be more convincing if the results could be reproduced for other human-centric datasets such as action recognition (UCF101) or audio-visual speech recognition (LRS3).

Due to the limitations in experimental validation, I recommend borderline reject.

The English is fine, but the paper is not easy to follow (perhaps due to the large number of components to explain).


Additional questions:

Does the authors have an explanation as to why V+A>P is so much worse than A>P in top1 accuracy in Table 3?

Isn't the pose just a processed form of the visual input? Is it just to make it more explicit so that it is easier for the network to process?

What information is fed into the network in the case of occlusion?

**Time Spent Reviewing:**

2hr

---

> ### Author Response · Authors · 2021-08-10
> **Response to Reviewer 8Z91**
>
> We would like to thank the reviewer for the valuable feedback. We address individual concerns below.
>
> * 1. Ablations to justify design choices.
>
> To justify the tri-modality design, we have conducted additional experiments by masking either the visual or pose modality and only using the remaining two modalities (i.e., audio/visual or audio/pose) for sound separation. The results are as follows:
>
> ---
>
> SDR: 7.82 &ensp;&emsp;&emsp;&emsp; SIR: 14.39  &emsp;&emsp;&emsp; SAR: 10.65 &emsp;&emsp;&emsp; (Masking is used for visual modality)
>
> SDR: 12.06  &emsp;&emsp;&emsp; SIR: 18.34 &emsp;&emsp;&emsp; SAR: 14.17 &emsp;&emsp;&emsp; (Masking is used for pose modality)
>
> SDR: 12.34 &emsp;&emsp;&emsp; SIR: 18.76 &emsp;&emsp;&emsp; SAR: 14.37 &emsp;&emsp;&emsp; (15% random masking for both visual and pose modality)
>
> ---
>
> Note, we do 15% random masking for both visual and pose modality by default as part of our original TriBERT model training. One can see that visual modality is contributing more, comparatively, than the pose modality, but both modalities are useful in improving the sound separation performance.
>
> * 2. Results for other human-centric datasets  such as action recognition (UCF101) or audio-visual speech recognition
>
> While we were unable to run experiments on UCF101 during the rebuttal, we have conducted additional experiments on instrument classification on the MUSIC21 dataset. We extract learned representations (i.e. audio / visual / pose features) from MUSIC21 dataset and use a simple MLP for instrument classification. We consider raw uni-modal features (which are input to TriBERT) as the baselines. We then compare these baselines to both uni-modal masked and multi-modal TriBERT features.
>
> ---
>
> Features    &emsp;&emsp; &emsp;&emsp;&emsp; &emsp;&emsp;&emsp; &emsp;&emsp;&emsp; &emsp;&emsp; &emsp; &emsp; &emsp; &emsp; &emsp; &emsp; &emsp;  &emsp; &emsp; &emsp; &emsp;     Top-1     &emsp;&emsp;&emsp;     Top-5     &emsp; &emsp;&emsp;     Top-10
>
> ---
>
> Audio feature w/o audio bert stream	   &emsp;&emsp; &emsp;&emsp;&emsp; &emsp;&emsp;&emsp; &emsp; &emsp; &emsp;&ensp;                56.87	 &emsp; &emsp;&emsp;              84.36	           &emsp; &emsp;&emsp;  93.84
>
> TriBERT audio feature with visual/pose masked   &emsp;&emsp; &emsp;&emsp;&emsp; &emsp;&ensp;   	 49.29         &emsp; &emsp;&emsp;               84.31          &emsp; &emsp;&emsp;              92.42
>
> TriBERT audio feature (our model)            &emsp; &emsp;&emsp;   &emsp; &emsp;&emsp;&nbsp;    &emsp; &emsp;&emsp;     &emsp; &emsp;&emsp;                          86.26         &emsp; &emsp;&emsp;               94.79           &emsp; &emsp;&emsp;             98.10
>
> ---
> Vision feature w/o visual bert stream            &emsp;&emsp;  &emsp; &emsp;&emsp; &emsp; &emsp;&emsp; &emsp; &emsp;&emsp;                  41.23       &emsp; &emsp;&emsp;             63.03          &emsp; &emsp;&emsp;            77.25
>
> TriBERT visual feature with  audio/pose  masked   &emsp; &emsp;&emsp; &emsp; &emsp;&emsp;&nbsp;     81.52      &emsp; &emsp;&emsp;              92.89         &emsp; &emsp;&emsp;             95.26
>
> TriBERT visual feature (our model)             &nbsp; &ensp; &emsp;&emsp; &emsp; &emsp;&emsp; &emsp; &emsp;&emsp; &emsp; &emsp;&emsp;                      87.20         &emsp; &emsp;&emsp;           94.31         &emsp; &emsp;&emsp;             96.68
>
> ---
>
> Pose feature w/o pose bert stream                &emsp; &emsp;&emsp; &emsp; &emsp;&emsp; &emsp; &emsp;&emsp; &emsp; &emsp;&emsp;                     39.81      &emsp; &emsp;&emsp;              76.78         &emsp; &emsp;&emsp;             91.47
>
> TriBERT pose feature with audio/visual masked     &emsp; &emsp;&emsp; &emsp; &emsp;&nbsp; &nbsp;&nbsp;    35.07          &emsp; &emsp;&emsp;          68.25          &emsp; &emsp;&emsp;            84.36
>
> TriBERT pose feature (our model)             &nbsp;&nbsp;&emsp; &emsp;&emsp; &emsp; &emsp;&emsp; &emsp; &emsp;&emsp; &emsp; &emsp;&emsp;                         87.68           &emsp; &emsp;&emsp;         96.21            &emsp; &emsp;&emsp;          98.58
>
> ---
>
> We can make a couple of observations here. First, audio and pose modalities appear to not benefit from within-modality transformer interactions of features. This makes sense since effectively we only have a single unique audio feature and pose features are highly redundant due to their long temporal context. The slightly lower performance over the “raw” input features is simply due to the learned co-attention across modalities (since in this experiment we do not fine-tune masked model variants). Notably, the visual features do tend to benefit from within-modality interactions among frames, because of the complementary and instantaneous nature of frame representation.
>
> Most importantly, one can see VERY significant improvements in classification when TriBERT is able to co-attend across modalities (our model). This is a strong indication of the benefits of multi-modal features produced by our TriBERT architecture. The improved performance and additional discriminative ability here can be attributed directly to cross-modal co-attention learned and induced by TriBERT.
>
> Additional experiments that we are currently working on, which we consider to be outside the scope of the NeurIPS paper, are discussed in response to Reviewer YXFS, Question 1.
>
> * 3. Why V+A -> P  is much worse than A>P in top1 accuracy in Table 3
>
> The performance of the two models is not necessarily directly comparable. Specifically, there are two issues that should be considered:
>
> - The input dimensionality and number of parameters of the V+A retrieval model is significantly larger, with an additional MLP layer used for fusion (see Line 354). This means that V+A model is more prone to overfitting. This is exhibited in slightly  lower performance for top1; notably top5 and top10 performance of V+A -> P is better.
>
> - The number of hard-negatives (n) is different in the two settings. For single-modality retrieval variants we use two distractors (negative pairings); while for two-modality variant we use three destractors. This may also marginally affect the performance, since in the n-way classification, having more distractors puts more focus on the negatives.
>
> However, we want to stress that the goal of these experiments isn’t to compare which modality or combination of modalities are best for retrieval. Instead, the goal is to illustrate the effectiveness of the TriBERT features. Each of the five retrieval models is simply an instance of a retrieval task. We can use any alternative (more sophisticated) models for retrieval here. The key observation is that in all five cases TriBERT features are SIGNIFICANTLY better in retrieval as compared with baseline features (used as input to TriBERT). This is strong evidence that TriBERT feature representations are effective. We make no claims with regards to optimality of the retrieval formulation or objective; it is simply used as a proxy for evaluating TriBERT features.
>
> * 4. Isn’t the pose just a processed form of the visual input?
>
> Yes, indeed. However, the pose is easier for the network to process and also contains more temporal context, so the information in the two streams is not completely redundant.  This is also supported by the ablation experiments conducted in response YXFS to Question 3c, where removing the pose reduces the quality of sound separation:
>
> ---
>
> SDR: 12.06 &emsp;&emsp;&emsp; SIR: 18.34 &emsp;&emsp;&emsp; SAR: 14.17 (Masking is used for pose modality)
>
> SDR: 12.34 &emsp;&emsp;&emsp; SIR: 18.76 &emsp;&emsp;&emsp; SAR: 14.37 (with both pose and visual modality)
>
> ---
>
> Similar observations were also made in  [2].
>
> [2] Gan, Chuang, et al. "Music gesture for visual sound separation." Proceedings of the IEEE/CVF Conference on Computer Vision and Pattern Recognition. 2020.
>
> * 5. What information is fed into the network in case of occlusion?
>
> There could be two forms of occlusion: partial-body occlusions or full-body occlusions. When the body is partially occluded, occluded joints are typically detected with low (close to 0) confidence. These low confidences are part of node representations for joints used by Spatial-Temporal Graph Convolutional Network to produce features for the keypoint (pose) representation stream of TriBERT. In other words, partial occlusions are actually encoded as part of features for the pose and could actually be useful (e.g., occlusion of the leg keypoints may indicate presence of cello or double bass). Full-body occlusions are generally rare, as this would indicate a person playing an instrument being fully occluded. In any case, we do not do anything “special” in such cases and simply obtain top K=2 human pose representations. If fewer than K human instances are present, we simply pad with zeros.

---

> > ### Comment · Reviewer_8Z91 · 2021-08-31
> > **Final comments**
> >
> > The rebuttal addressed many of my concerns on experimental validation. I will change my rating from 5 to 6.

---

### Official Review · Reviewer_WFQN · 2021-07-19

**Rating:** 6
**Confidence:** 4

**Summary:**

By pointing out the recent transformer-based models are mostly designed for visual-language data, this paper introduce TriBERT which specifically targeting on audio-visual modalities to address such limitation. Inspired by ViLBERT, TriBERT is a transformer based model which enables contextual feature learning across three modalities: vision, pose and audio with the use of flexible co-attention. Besides, this paper introduce a learned visual tokenization scheme based on spatial attention and leverage weak-supervision to allow granular cross-modal interactions for visual and pose modalities. To demonstrate the proposed approach, TriBERT is pretrained on MUSIC221 and show improved performance in audio-visual sound source separation and other audio-visual-pose retrieval.

**Limitations And Societal Impact:**

Some of my suggestions:
1) add more SOTA approaches for comparison on sound source localization task
2) add SOTA comparisons on retrieval task
3) add ablation studies of the proposed approach, e.g. comparisons of different visual tokenization approaches
4) further demonstrate the claim of 'learning generic audio-visual representations', e.g. consider tasks of audio/video classification
5) comparing with other audio-visual work, pose is the an extra modalities and the unique part in this paper. I would suggest to add some experiments and analysis on the contribution of adding pose
6) no code was submitted. I am wondering the reproducibility of this work
7) minor: alter positions of  Table 1 & 2

**Main Review:**

This paper introduces TriBERT, a three-stream model which can accept three modalities, i.e. audio, visual, pose. By utilizing tri-modal co-attention blocks, generic audio-visual representations are achieved for audio-visual tasks of sound source separation and audio-visual-pose retrieval. This work of extending the usage of transformer-based models for more modalities other than just visual and language, i.e. audio-visual, is well motivated.

However, my main concern is the novelty of this work is limited. Though, to extending transformer to audio and pose is not a trivial problem, the approach been used is straightforward. The whole TriBERT model is kind of a combination of some existing building blocks, e.g. BERT, graph CNN, segmentation net, Vggish net, UNet, etc. The training objective is designed as a multi-task manner, i.e. visual and pose classification, audio classification, and multi-modal sound source separation. From both aspects of model design, and algorithm/training objective, I do see much novelty appears.
Besides, the experiments are not strong enough. First, with comparing with only a few SOTA approaches,  I can not be fully convinced. Also, why there no comparisons with sota for retrieval task?
For many recent audio-visual representation learning work, the approaches are usually demonstrated on 2-3 audio-visual tasks, e.g. audio event classification, and activity classification. I am wondering would these tasks can be considered to demonstrate the authors' claim of 'learning generic audio-visual representation'?
I also did not see any ablation studies of the proposed approach. For example, the authors claim they proposed a new learned visual tokenization scheme, but no experimental results are shown to demonstrate the effectiveness of this tokenization scheme.


**Time Spent Reviewing:**

3 hrs

---

> ### Author Response · Authors · 2021-08-10
> **Response to Reviewer WFQN**
>
> We would like to thank the reviewer for the valuable feedback. We address individual concerns below.
>
> * 1. Add more SOTA approaches for comparison on sound source localization.
>
> To the best of our knowledge, prior SOTA approaches did not provide quantitative results for the sound source localization. In fact, MUSIC21 and MUSIC datasets do not have annotations for the sounding objects. As such, we are not able to provide quantitative comparison. For the camera ready, we will attempt to annotate such objects ourselves and provide a comparative evaluation. This was not possible in time for the rebuttal.
>
>
> * 2. Add SOTA comparisons on retrieval tasks.
>
> There are many standard cross-modal retrieval models, but none to our knowledge that are specifically designed for audio-pose-visual tasks. As such, we would need to adapt existing models to the task at hand to compare. Further, the key to our approach is the feature learning framework. Extracted TriBERT features can be subsequently used with ANY retrieval network. Hence the value of comparison is to see how much retrieval improves, given proposed features, under identical retrieval pipeline and objective; this was the focus of our experiments. We would be happy to add similar TriBERT experiments for any retrieval model suggested by the reviewer.
>
>
> * 3. Comparison of different visual tokenization approaches.
>
> This is a very good idea. We have started working on these experiments, but given the training time (9 days for pre-training, 3 days for fine-tuning) it is not possible to obtain such a comparison for the rebuttal.
>
>
> * 4. To further demonstrate the claims … consider saks of audio/video classification
>
> We conducted additional experiments to show effectiveness of learned generic representations as suggested by the reviewer (thank you). We extract learned TriBERT representations (i.e. audio / visual / pose features) from the MUSIC21 dataset and use a simple MLP for instrument classification. We consider raw uni-modal features (which are input to TriBERT) as the baselines. We then compare these baselines to both uni-modal masked and multi-modal TriBERT features. In doing so, no fine-tuning is employed (this can further improve results).
>
> ---
>
> Features    &emsp;&emsp; &emsp;&emsp;&emsp; &emsp;&emsp;&emsp; &emsp;&emsp;&emsp; &emsp;&emsp; &emsp; &emsp; &emsp; &emsp; &emsp; &emsp; &emsp;  &emsp; &emsp; &emsp; &emsp;     Top-1     &emsp;&emsp;&emsp;     Top-5     &emsp; &emsp;&emsp;     Top-10
>
> ---
>
> Audio feature w/o audio bert stream	   &emsp;&emsp; &emsp;&emsp;&emsp; &emsp;&emsp;&emsp; &emsp; &emsp; &emsp;&ensp;                56.87	 &emsp; &emsp;&emsp;              84.36	           &emsp; &emsp;&emsp;  93.84
>
> TriBERT audio feature with visual/pose masked   &emsp;&emsp; &emsp;&emsp;&emsp; &emsp;&ensp;   	 49.29         &emsp; &emsp;&emsp;               84.31          &emsp; &emsp;&emsp;              92.42
>
> TriBERT audio feature (our model)            &emsp; &emsp;&emsp;   &emsp; &emsp;&emsp;&nbsp;    &emsp; &emsp;&emsp;     &emsp; &emsp;&emsp;                          86.26         &emsp; &emsp;&emsp;               94.79           &emsp; &emsp;&emsp;             98.10
>
> ---
> Vision feature w/o visual bert stream            &emsp;&emsp;  &emsp; &emsp;&emsp; &emsp; &emsp;&emsp; &emsp; &emsp;&emsp;                  41.23       &emsp; &emsp;&emsp;             63.03          &emsp; &emsp;&emsp;            77.25
>
> TriBERT visual feature with  audio/pose  masked   &emsp; &emsp;&emsp; &emsp; &emsp;&emsp;&nbsp;     81.52      &emsp; &emsp;&emsp;              92.89         &emsp; &emsp;&emsp;             95.26
>
> TriBERT visual feature (our model)             &nbsp; &ensp; &emsp;&emsp; &emsp; &emsp;&emsp; &emsp; &emsp;&emsp; &emsp; &emsp;&emsp;                      87.20         &emsp; &emsp;&emsp;           94.31         &emsp; &emsp;&emsp;             96.68
>
> ---
>
> Pose feature w/o pose bert stream                &emsp; &emsp;&emsp; &emsp; &emsp;&emsp; &emsp; &emsp;&emsp; &emsp; &emsp;&emsp;                     39.81      &emsp; &emsp;&emsp;              76.78         &emsp; &emsp;&emsp;             91.47
>
> TriBERT pose feature with audio/visual masked     &emsp; &emsp;&emsp; &emsp; &emsp;&nbsp; &nbsp;&nbsp;    35.07          &emsp; &emsp;&emsp;          68.25          &emsp; &emsp;&emsp;            84.36
>
> TriBERT pose feature (our model)             &nbsp;&nbsp;&emsp; &emsp;&emsp; &emsp; &emsp;&emsp; &emsp; &emsp;&emsp; &emsp; &emsp;&emsp;                         87.68           &emsp; &emsp;&emsp;         96.21            &emsp; &emsp;&emsp;          98.58
>
> ---
>
> We can make a couple of observations here. First, audio and pose modalities appear to not benefit from within-modality transformer interactions of features. This makes sense since effectively we only have a single unique audio feature and pose features are highly redundant due to their long temporal context. The slightly lower performance over the “raw” input features is simply due to the learned co-attention across modalities (since in this experiment we do not fine-tune masked model variants). Notably, the visual features do tend to benefit from within-modality interactions among frames, because of the complementary and instantaneous nature of visual representation.
>
> Most importantly, one can see VERY significant improvements in classification when TriBERT is able to co-attend across modalities (our model). This is a strong indication of the benefits of multi-modal features produced by our TriBERT architecture. The improved performance and additional discriminative ability here can be attributed directly to cross-modal co-attention learned and induced by TriBERT.
>
>
> * 5. Add some experiments and analysis on the contribution of adding pose
>
> We conduct additional experiments by masking visual/pose modality during fine-tuning on the MUSIC dataset. Masking is equivalent to erasing the corresponding modality. The result of this ablation are as follows:
>
> ---
>
> SDR: 7.82 &emsp;&emsp;&emsp;&ensp; SIR: 14.39 &emsp;&emsp;&emsp; SAR: 10.65 &emsp;&emsp;&emsp; (Masking is used for visual modality)
>
> SDR: 12.06 &emsp;&emsp;&emsp; SIR: 18.34 &emsp;&emsp;&emsp; SAR: 14.17 &emsp;&emsp;&emsp; (Masking is used for pose modality)
>
> SDR: 12.34 &emsp;&emsp;&emsp; SIR: 18.76 &emsp;&emsp;&emsp; SAR: 14.37 &emsp;&emsp;&emsp; (15% random masking for both visual and pose modality)
>
> ---
>
> We do 15% random masking for both visual and pose modality by default as part of our TriBERT model training.
>
> Overall, one can see that visual modality is contributing more, comparatively, than the pose modality, but both modalities are useful in improving the performance. When pose modality is used alone (top row) the performance is still reasonable. This suggests some level of redundancy in pose and visual information, rather than lack of information in pose modality.
>
>
> * 6. Reproducibility of the work
>
> We will release code and all pre-trained models to the public upon acceptance of the paper (please see Footnote 2, Page 7).
>
>
> * 7. Minor: Alter positions of Table 1 & 2
>
> This was an artifact of LaTex and us trying to squeeze the paper into the page limit. We will address this in the revision.

---

> > ### Comment · Reviewer_WFQN · 2021-08-31
> > **Final comments**
> >
> > Thanks for the detailed response. It addressed most of my concerns, so I would like to raise my score to 6. While I hope the authors will complete the promised experiments, and add the new numbers in the final version (including experiments suggested by other reviewers).

---

### Official Review · Reviewer_YXFS · 2021-07-19

**Rating:** 5
**Confidence:** 5

**Summary:**

This work proposes TriBERT to handle the modalities of RGB image, poses and audios for tackling the task of sound source separation. The BERT model is used to extract features following ViLBERT. Experiments show appealing results on sound separation metrics compared with the author-reproduced previous SOTA methods.

**Limitations And Societal Impact:**

The authors have addressed their limitations. But these are major weaknesses of their work that could not be simply regarded as limitations. The authors should consider changing their story.

Video results should be included to show the qualitative results, so that more limitations could be revealed

**Main Review:**

This paper illustrates a relatively reasonable story in its introduction, however, the story cannot be fully supported by the experiments conducted. The usage of the BERT model in the field of visual sound separation is quite interesting. The model descriptions are mostly clear enough for reproducibility. The reviewer thinks this paper is of certain value to the field of visual sound separation.

** The strengths of this paper are summarized as follows:

++ This paper is well written and easy to follow.

++ The idea of involving BERT in visual sound separation is interesting.

++ Most of the model design is reasonable and with care, which is important for application papers.

** The weaknesses of the paper are as follows:

1. The conducted experiments do not match with the story in the introduction and the title.

a) The title is “Human-centric” audio-visual representation learning, however, the experiments are conducted on MUSIC21 dataset with sound source separation task. I am not saying that separating instrument sound is not entirely “human-centric”, but this is only a relatively small area in the  “human-centric” audio-visual field. I am expecting to see experiments conducted on human speech and its relationship with maybe co-speech gestures when I read the introduction. The experiments cannot support the title or the introduction at all.

b) The core of ViLBERT or related work, is to use proxy tasks for training and show the effectiveness of learned features with downstream applications, while this paper basically focuses on sound separation. Although this is easy to understand: there are not so many downstream tasks if the model is trained on MUSIC dataset.

The reviewer suggests that the authors either focus on visual sound separation (with a title like “TriBERT for visual sound separation”) or conduct experiments on large-scale “human-centric” datasets such as TED talks or AudioSet, and show the effectiveness of representation learning on more downstream tasks.

2. No qualitative results and no video supplementary shown. Demo videos are extremely important for audio-visual papers. With no video results, it is impossible for reviewers to know the true performance of the method.

Also, the authors do not provide any visual results of visual segmentation while the visual branch is designed for this.
The retrieval experiments are weird. Though I understand the authors are trying to match the “representation learning” in the title, these kinds of experiments are useless and have never been conducted in related studies.

3. The quantitative experiments are not persuasive. Although our community cares less about computational cost, the paper lacks a lot of ablation studies. For example, this paper leverages attention U-Net as backbone, which is rarely used in previous works. The entire visual network is more complicated than all previous work. And the three modalities in the TriBERT are not evaluated efficiently. What if one of the modality is erased? More needed ablations are not listed.

4. Details. Why use only three frames which is the same as PixelPlayer for the pose branch? This is not reasonable given that the pose frames should cover more temporal information as performed in Music Gesture.

Why Table1 is below Table2 and why evaluate on MUSIC alone given that it is a subset of MUSIC21? The settings are not reasonable.

-------------------------------------

The rebuttal addresses some of my concerns and provides positive revision directions. However, it requires a major revision from the current form (the story needs to change, or large amounts of experiments should be shown), thus I still do not recommend acceptance of this paper. I raise my rating from 4 to 5 and encourage the authors to keep perfecting this paper.

**Time Spent Reviewing:**

6 hours

---

> ### Author Response · Authors · 2021-08-10
> **Response to Reviewer YXFS**
>
> We would like to thank the reviewer for the valuable feedback. We address individual concerns below.
>
> * 1. The conducted experiments do not match with the story in the introduction and the title.
>
> We propose a model for learning a joint representation across visual, (keypoint) pose and audio features. To our knowledge, this is the first attempt to learn such a tri-modal representation using BERT-like architecture. Fundamentally, our formulation is general and should be applicable to a variety of problem domains and settings with minimal to no modifications. This is the intuition we tried to convey in the title and the introduction. We apologize if this came off as overclaiming.
>
> We completely agree that we do not test the effectiveness of TriBERT across all possible datasets and problem domains where it may in fact be useful. Since the submission of the paper to NeurIPS, we have started training TriBERT with AudioSet for sound separation and VoxCeleb2 for speech recognition. These experiments are on-going and our plan is to include them in the future journal version of the current paper.
>
> In the meantime, we would be happy to adjust the title and introduction following the suggestion by the reviewer to better reflect the narrower scope of NeurIPS experiments. For example: “TriBERT: Full-body Human-centric Audio-visual Representation Learning for Visual Sound Separation”.
>
>
> * 2. No qualitative results and no video supplementary shown.
>
> We apologize for failing to include videos. We realize this was an oversight on our part. Due to the nature of OpenReview, we can not upload these results as part of the rebuttal without including external links (which would violate the blind nature of the review process). For what it is worth, we can assure the reviewer that the differences in the audio between the baselines and our model are discernible. Upon acceptance of the paper, we will release the corresponding videos on our project page.
>
>
> * 3a. Computational cost of attention U-Net
>
> Attention U-Net uses a simple additive soft attention to highlight relevant regions of the audio spectrograms. The overhead of attention U-Net over U-Net is fairly minimal. Specifically, in terms of the number of parameters, attention U-Net contains a modest ~9% more parameters as compared to U-Net and the inference speed is only ~7% slower. The training time is approximately the same between the two models in our experience. The computational cost of attention U-Net is analyzed in detail in Table 1 of [1]. We will add discussion of this to the paper.
>
> [1] Oktay, Ozan, et al. "Attention u-net: Learning where to look for the pancreas." arXiv preprint arXiv:1804.03999 (2018).
>
>
> * 3b. Visual network is more complicated than all previous work
>
> We respectfully disagree. Our visual network uses a similar pipeline to previous works and the same ResNet backbone. The key difference is that instead of using a pre-trained object detector, we use an end-to-end network to detect objects using attention. Doing so reduces our reliance on an external object detection scheme, that itself needs to be trained, and reduces pre-processing. Further, we note that our visual network is modeled after, and is identical to, the one introduced in [42] (to which we directly compare in Table 1).
>
>
> * 3c. What if one of the modality is erased? More needed ablations are not listed
>
> Thank you for your suggestion. We conducted additional experiments by masking visual/pose modality during fine-tuning on the MUSIC dataset. Masking is equivalent to erasing a modality. The results of this ablation are as follows:
>
> ---
>
> SDR: 7.82 &emsp;&emsp;&emsp;&ensp; SIR: 14.39 &emsp;&emsp;&emsp; SAR: 10.65 &emsp;&emsp;&emsp; (Masking is used for visual modality)
>
> SDR: 12.06 &emsp;&emsp;&emsp; SIR: 18.34 &emsp;&emsp;&emsp; SAR: 14.17 &emsp;&emsp;&emsp; (Masking is used for pose modality)
>
> SDR: 12.34 &emsp;&emsp;&emsp; SIR: 18.76 &emsp;&emsp;&emsp; SAR: 14.37 &emsp;&emsp;&emsp; (15% random masking for both visual and pose modality)
>
> ---
>
> In this paper, we are using sound separation as our primary task. Therefore, we do not consider masking for the audio modality. We do 15% random masking for both visual and pose modality by default as part of our TriBERT model training.
>
> Overall, one can see that visual modality is contributing more, comparatively, than the pose modality, but both modalities are useful in improving the performance.
>
>
> * 4a. Why use only three frames which is the same as a PixelPlayer for a pose branch?
>
> BERT-based architectures, including ours, require large GPU memory and longer training time. Therefore, we use only three frames to reduce computational cost, but the number of frames can be easily increased with the same architecture (if resources allow).
>
> Further, we would like to highlight that a pose feature for one frame, actually takes into account T=256 frames (see Line 198) of poses using a Spatial-Temporal Graph Convolutional Network (see Line 191). Therefore long-term contextual pose information is taken into account. For more details please see [52].
>
>
> * 4b. Why Table 1 below Table 2?
>
> Apologies, this was an artifact of LaTex and us trying to squeeze the paper into the page limit. We will address this in the revision.
>
>
> * 4c. Why evaluate on MUSIC alone given that it is a subset of MUSIC21
>
> Even though the MUSIC dataset is a subset of MUSIC21, this subset has notably different statistical characteristics (different number of classes, relative distribution of data, etc.). As a result of this, a model trained on MUSIC21 is not necessarily going to produce optimal performance on the MUSIC dataset. To illustrate this we report sound separation performance on MUSIC dataset before and after fine-tuning:
>
> ---
>
> Before fine-tuning &nbsp;&nbsp;&nbsp;&nbsp;&nbsp;&nbsp; SDR: 11.81 &nbsp;&nbsp;&nbsp;&nbsp;&nbsp;&nbsp; SIR: 17.53 &nbsp;&nbsp;&nbsp;&nbsp;&nbsp;&nbsp; SAR: 14.69
>
> ---
>
> After fine-tuning &nbsp;&nbsp;&nbsp;&nbsp;&nbsp;&nbsp;&nbsp;&nbsp;&nbsp; SDR: 12.34 &nbsp;&nbsp;&nbsp;&nbsp;&nbsp;&nbsp; SIR: 18.49 &nbsp;&nbsp;&nbsp;&nbsp;&nbsp;&nbsp; SAR: 14.63
>
> ---
>
> We can see that after fine-tuning, sound separation results have improved in terms of SDR and SIR.  SAR is responsible for capturing absence of artifacts and it can be higher even though separation results are poor. Note that if MUSIC21 and MUSIC had similar statistics and the model trained on MUSIC21 was “sufficient”, the performance with fine-tuning would not improve. This is not the case. Therefore MUSIC dataset can be considered to be a “down-the-stream” dataset on which effectiveness of features, using fine-tuning, can be tested.

---

### Comment · Area_Chair_S2RM · 2021-08-18
**Post-rebuttal discussion!**

Dear reviewers:

Thanks so much for spending time reviewing this paper.

Could you please spend time reading the author's response and share your opinion on this work?

Thanks
AC

---

> ### Comment · Reviewer_voNw · 2021-08-24
>
> Hello,
>
> I decided to keep my initial evaluation ("6: marginally above the acceptance threshold"), as I believe that the paper has obvious merits, it is well-written, and the authors have done a very good job in answering the rebuttal questions.
>
> At the same time, I also agree with the concerns of the reviewer YXFS about the fact that a significant number of experiments (reported and promised in rebuttal) should be added to the final version of the paper.

---

### Decision · Program_Chairs · 2021-09-27

**Decision:**

Accept (Poster)

**Comment:**

This work proposes a new transformer-based model that can learn across three modalities: vision, pose, and audio for tackling the task of sound source separation. The authors did a good job during rebuttal and turned two slightly negative reviewers into positive ones.  The final score is very borderline (three borderline accepts and one borderline reject).  This work could clearly benefit from a more extensive experimental evaluation, but AC feels this work is very interesting and deserves to be published on NeurIPS. The reviewers did raise some valuable concerns that should be addressed in the final camera-ready version of the paper. The authors are also encouraged to make other necessary changes.